# The Usefulness of In Vitro Percutaneous Absorption Experiments Applying the Infinite Dose Technique to Predict In Vivo Plasma Levels: Comparison of Model-Predicted and Observed Plasma Concentrations of Nortriptyline in Rats

**DOI:** 10.3390/pharmaceutics14071457

**Published:** 2022-07-12

**Authors:** Iris Usach, Sara Di Marco, Octavio Díez, Manuel Alós, José-Esteban Peris

**Affiliations:** Department of Pharmacy and Pharmaceutical Technology and Parasitology, University of Valencia, Burjassot, 46100 Valencia, Spain; iris.usach@uv.es (I.U.); sara_di_marco@virgilio.it (S.D.M.); octavio.diez@uv.es (O.D.); manuel.alos@uv.es (M.A.)

**Keywords:** nortriptyline, percutaneous absorption, model predicted concentrations, infinite dose, finite dose

## Abstract

The aims of this study were to evaluate the feasibility of a nortriptyline (NT) formulation for transdermal administration and to assess the usefulness of an estimated kinetic parameter (*k_out_*) using the in vitro infinite dose technique to predict in vivo plasma levels when used in combination with pharmacokinetic parameters. To do so, a simple one-compartment model was used to describe the transport of a permeant across a membrane (skin). This model provides relatively simple expressions for the amount of permeant in the skin, the cumulative amount of permeant that crosses the skin, and the flux of permeant, for both the infinite and the finite dose regimens. Transdermal administration of the formulated NT gel to rats resulted in plasma levels of approximately 150 ng/mL between 8 and 30 h post-administration. These levels were higher than the minimum concentration of 40 ng/mL recommended for smoking cessation therapy and slightly higher than the upper limit of the therapeutic range for the treatment of depression in humans. The one-compartment model used to describe transport across the skin was connected to a two-compartment pharmacokinetic model used to predict NT plasma concentrations in rats using the *k_out_* determined in vitro and the values of other pharmacokinetic parameters obtained in vivo. The predicted concentrations were close to the observed plasma levels and the time profiles were similar for both types of data. These results show the usefulness of the *k_out_* parameter determined in vitro to predict plasma concentrations of drugs administered percutaneously.

## 1. Introduction

The transdermal route of drug administration has received a great deal of attention from researchers and drug manufacturing companies in recent years [1]. Transdermal drug delivery (TDD) allows the obtention of systemic effects with additional advantages when compared to the oral or the parenteral administration routes [2,3]. In contrast to the oral route, transdermal drug administration obviates gastrointestinal absorption, as well as hepatic first-pass metabolism, and it minimises the adverse effects brought about by peak plasma drug concentrations and improves patient compliance. Moreover, it provides a non-invasive alternative to parenteral injections, entails no risk of infection, and provides constant blood levels for drugs with a narrow therapeutic window [4]. However, a big limitation of TDD is the skin’s intrinsic low permeability, which restricts drugs amenable to administration by this route. The evaluation of percutaneous permeation is key to knowing if a given drug could be a candidate for TDD, as well as if it can be performed using in vivo or in vitro approaches. The most appropriate approach is the use of in vivo studies in volunteers (assuming that the drugs assessed are destined for use in humans). However, there are ethical and economic reasons that limit the use of these assays in humans, and several alternative in vitro methods, which use human skin obtained from corpses or plastic surgery, cultured skin, animal skin, or artificial membranes, have been developed [5].

In vitro studies measure the diffusion of the chemical of interest, or permeant, into and across the skin. To carry out these studies, Franz diffusion cells with two chambers separated by a piece of skin are usually employed. The formulation containing the permeant is introduced into the donor chamber and the concentration of the permeant in the fluid that fills the receptor chamber is measured at different sampling times. For each sampling time, the cumulative amount of permeant per unit area diffused across the skin, or amount at the receptor chamber per unit area (*Q_R_*), can be calculated considering the measured concentration, the volume of fluid in the receptor chamber, and the exposed surface of the skin.

Two approaches, depending on the dose of the permeant, can be used for the study of percutaneous permeation: an infinite or a finite dose regimen. An infinite dose regimen implies that the permeant concentration in contact with the skin remains constant throughout the study, while in a finite dose regimen, this concentration decreases with time.

When studying percutaneous permeation in vitro using diffusion cells, the use of an infinite dose regimen provides experimental data (*Q_R_*) that show a curved profile when plotted vs. time (Figure 1A). This profile can be described by an analytical solution of Fick’s second law of diffusion, under the assumption that the skin barrier acts as a pseudo-homogeneous membrane [6,7]:(1)$$Q_{R}=K_{p}C_{D}t-K_{p}C_{D}T_{lag}-\frac{12K_{p}C_{D}T_{lag}}{\pi^{2}}\overset{\infty }{\underset{n=1}{\sum }}\frac{{\left(-1\right)}^{n}}{n^{2}}e^{\left(\frac{-n^{2}\pi^{2}t}{6T_{lag}}\right)}$$
being
(2)$$K_{p}=\frac{KD}{h}$$
(3)$$T_{lag}=\frac{h^{2}}{6D}$$

In the above equations, *K_p_* is the permeability coefficient, *C_D_* is the concentration of permeant at the donor chamber, *t* is the time elapsed from the beginning of the experiment, *T_lag_* is the lag time, *K* is the partition coefficient of the permeant between the outmost layer of the skin and the vehicle of the formulation applied at the donor chamber, *D* is the diffusion coefficient, and *h* is the thickness of the skin or membrane. The curve in Figure 1A tends to a straight line at some time after the beginning of the experiment, and the appearance of such a pseudo-straight line is interpreted as a pseudo-steady state in the diffusion process. When considering only the data (*Q_R_*, *t*) obtained at that pseudo-steady state, Equation (1) can be simplified as follows:(4)$$Q_{R}=K_{p}C_{D}t-K_{p}C_{D}T_{lag}$$

In this equation, the slope represents the steady-state flux (*J_ss_*):(5)$$J_{ss}=K_{p}C_{D}$$
and the *t*-axis intercept is the *T_lag_* value (Figure 1A). Equation (5) is often employed to determine the value of *K_p_*, considering the value of the slope (*J_ss_*) obtained by linear regression according to Equation (4) and the constant concentration of the permeant in the donor chamber.

Figure 1B shows that the flux (J) increases with time until reaching a pseudo-constant value (*J_ss_*), while Figure 1C displays the expected plasma concentration profile of a permeant considering an infinite dose regimen or constant concentration of permeant on the skin along the study. If the infinite dose regimen is maintained for a relatively long time, a steady-state plasma concentration (*C_ss_*) will be reached.

In the case of an in vitro finite dose regimen study (Figure 1D), the equation of *Q_R_* vs. time derived from Fick’s second law of diffusion is much more complex than Equation (1) [7], a maximum flux (*J_max_*) at time *T_Jmax_* (Figure 1E) is achieved (as opposed to a steady state flux), and the expected plasma concentration profile also shows a peak value (*C_max_*) at some time (*T_Cmax_*) after the administration (Figure 1F). 

The infinite dose regimen is employed more often in in vitro studies than the finite dose regimen because of its less complex mathematical treatment, its obtention of parameters, such as *J_ss_*, *K_p_*, and *T_lag_*, with a relatively simple interpretation, and the possibility of predicting a *C_ss_* value using the following equation: (6)$$C_{ss}=\frac{J_{ss}S}{CL}=\frac{K_{p}C_{D}S}{CL}$$
where *S* is the surface of the skin exposed to the formulation and *CL* is the plasma clearance of the permeant. However, the in vivo scenario is often closer to a finite dose than an infinite dose regimen, with plasma concentration curves similar to that in Figure 1F [8,9,10,11].

In addition to the diffusion models, compartmental pharmacokinetic models and physiologically based pharmacokinetic (PBPK) models have been used to describe the in vivo percutaneous absorption of different substances [12,13,14].

Nortriptyline (NT) is a tricyclic antidepressant drug that belongs to the class of non-selective monoamine uptake inhibitors. It is widely used in the treatment of unipolar depression [15] and there is growing evidence of its efficacy in pharmacological smoking cessation therapy [16,17,18]. The therapeutic range of NT for the treatment of depression is 50–150 ng/mL [19,20]; however, the concentrations required for smoking cessation therapy are lower (≥40 ng/mL) [21].

The oral administration of NT is associated with some disadvantages, such as low oral bioavailability (between 30 and 50%) and side effects associated with plasma level fluctuations [22,23]. Hence, the development of a transdermal delivery system has many advantages: high patient compliance, minimum adverse effects, constant plasma levels (not fluctuating), and avoiding the first pass effect and the variability associated with the gastrointestinal tract [24]. NT is classified as group I (high solubility and high permeability) in the Biopharmaceutical Classification System (BCS) [25].

The objectives of this study were to evaluate the feasibility of an NT formulation for transdermal administration and to assess the usefulness of the in vitro infinite dose technique to predict in vivo plasma levels of NT in rats percutaneously administered with the aforementioned formulation. To enable this, the experimental data obtained in vitro were analysed using a compartmental model to obtain the value of a kinetic constant for the diffusion of the drug from the skin to the receptor chamber of the diffusion cell. The value of this constant was combined with the pharmacokinetic parameters of NT obtained in rats to predict plasma concentrations, which were compared with those obtained after the in vivo administration of the formulation.

## 2. Materials and Methods

### 2.1. Theory: One-Compartmental Model-Derived Equations

A simple one-compartment model can be used to describe the transport of a permeant across a membrane (skin) that separates the donor from the receptor chambers in a diffusion cell (Figure 2).

In this model, the concentration of the permeant is assumed to be homogeneous inside the compartment (membrane or skin), the entrance of permeant into the compartment from the donor is described by means of zero-order kinetics (constant-rate input) in the case of an infinite dose experiment (Figure 2A) or first-order kinetics in the case of a finite dose experiment (Figure 2D), and the exit of the permeant from the compartment to the receptor is described by first-order kinetics for types experiments. 

#### 2.1.1. Infinite Dose

The differential equation describing the change in the amount of permeant in the membrane with time when the permeant enters into the membrane at a constant rate is as follows:(7)$$\frac{dQ_{M}}{dt}=R_{0}-K_{out}Q_{M}$$
where *Q_M_* is the amount of permeant in the membrane per unit area, *R*_0_ is the constant rate per unit area at which the permeant enters the membrane, and *K_out_* is the first-order rate constant characterizing the permeant output from the membrane to the receptor chamber.

An equation describing the time course of the amount of permeant in the membrane can be obtain by integrating Equation (7) [26]:(8)$$Q_{M}=\frac{R_{0}}{k_{out}}\left(1-e^{k_{out}t}\right)$$

A mass balance considering the cumulative amount of permeant that enters the membrane (*R*_0_ × *t*), the amount of permeant in the membrane (obtained with Equation (8)), and the cumulative amount in the receptor (*Q_R_* = *R*_0_*t* − *Q_M_*) provides the following equation:(9)$$Q_{R}=R_{0}t-\frac{R_{0}}{k_{out}}\left(1-e^{k_{out}t}\right)$$

Equation (9), which is simpler than Equation (1), can be fitted to the data (*Q_R_*, *t*) obtained in an infinite dose experiment in order to estimate the values of *R*_0_ and *k_out_*.

When considering only the data (*Q_R_*, *t*) obtained at the pseudo-steady state, Equation (9) can be simplified as a straight line:(10)$$Q_{R}=R_{0}t-\frac{R_{0}}{k_{out}}$$
where the slope is the value of *R*_0_ and the *t*-axis intercept is the value of 1/*k_out_* (Figure 2B). In the above equations, *R*_0_ also represents the steady-state flux (Figure 2C) and can be used as a replacement of *J_ss_* in Equation (5) in order to estimate the value of *K_p_* as the ratio of *R*_0_ to *C_D_*. Additionally, the inverse of *k_out_* can be interpreted in a similar way as *T_lag_* in Equations (1)–(4). *T_lag_* is related to the time to reach a certain fraction of steady state. In this way, the flux (*J*) after 2.7 times the *T_lag_* value has been estimated in about 97.5% of *J_ss_* [7]. The following equation gives *J* as a function of time according to the proposed model (Figure 1A):(11)$$J=R_{0}\left(1-e^{k_{out}t}\right)$$

This equation can be used to deduce that the time required to obtain 97.5% of *J_ss_* is 3.7 times the 1/*k_out_* value.

#### 2.1.2. Finite Dose

First-order kinetics are considered in the model for both the input of permeant into the membrane and its output to the receptor chamber (Figure 2D). The set of differential equations describing the model is as follows:(12)$$\frac{dQ_{D}}{dt}=-k_{in}Q_{D}$$
(13)$$\frac{dQ_{M}}{dt}=k_{in}Q_{D}-k_{out}Q_{M}$$
where *k_in_* is the first-order rate constant for permeant input into the membrane. By integrating these equations, the following equations describing the time course of *Q_D_* and *Q_M_* can be obtained [26]:(14)$$Q_{D}=Q_{0}e^{-k_{in}t}$$
(15)$$Q_{M}=\frac{k_{in}Q_{0}}{\left(k_{in}-k_{out}\right)}e^{-k_{out}t}+\frac{k_{in}Q_{0}}{\left(k_{out}-k_{in}\right)}e^{-k_{in}t}$$
where *Q*_0_ is the dose, or initial amount, of permeant per unit area in the donor chamber. A mass balance considering the initial amount of permeant and the amounts at the donor chamber, membrane, and receptor chamber (*Q_R_* = *Q*_0_ − *Q_D_* − *Q_M_*) yields:(16)$$Q_{R}=k_{in}Q_{0}\left[\frac{1}{k_{in}}+\frac{e^{-k_{out}t}}{\left(k_{out}-k_{in}\right)}+\frac{k_{out}e^{-k_{in}t}}{k_{in}\left(k_{in}-k_{out}\right)}\right]$$

The cumulative curve described by Equation (16) is shown in Figure 2E. Additionally, the following equation gives the flux as a function of time (Figure 2F):(17)$$J=\frac{k_{out}k_{in}Q_{0}}{\left(k_{in}-k_{out}\right)}e^{-k_{out}t}+\frac{k_{out}k_{in}Q_{0}}{\left(k_{out}-k_{in}\right)}e^{-k_{in}t}$$

Equations (16) and (17) can be fitted to the experimental data (*Q_R_*, *t*) or (*J*, *t*) in order to obtain the values of *k_in_* and *k_out_*.

By making the first derivative of Equation (17) equal to zero, the following equation can be obtained for the time to achieve *J_max_*:(18)$$T_{J_{max}}=\frac{\ln\left(k_{out}/k_{in}\right)}{\left(k_{out}-k_{in}\right)}$$
and the value of *J_max_* can be obtained by replacing *t* in Equation (17) with the *T_jmax_* value obtained with Equation (18).

In both kinds of experiments—infinite and finite dose—*k_out_* represents the same constant, which means that its value can be obtained with an infinite dose experiment and used later for simulations considering a finite dose administration, as described below.

### 2.2. Chemicals

Nortriptyline hydrochloride (NT), hydroxypropyl methyl cellulose (HPMC), tween^®^ 80 (T80), oleic acid (OA), propylene glycol (PG), and 9-fluorenylmethyl chloroformate (Fmoc-Cl) where purchased from Sigma-Aldrich (Madrid, Spain). All other reagents were of analytical or HPLC grade.

### 2.3. Animals

Male Wistar rats (280–310 g) were used in this study, and the experiments were performed in accordance with the authors’ institutional rules for the use of animals in research. All animals were housed under standard conditions and had ad libitum access to water and a standard laboratory diet. Protocols for the in vivo studies using rats were approved by the Ethics Committee for Experimentation and Animal Welfare (University of Valencia), approval code A1352991914316 (3 May 2013).

### 2.4. Preparation of NT Gel

The composition of the NT gel was (m/m): water (34%), PG (30%), ethanol (25%), NT (5%), OA (2%), T80 (2%), and HPMC (2%). For the gel preparation, HPMC was dispersed in water with the help of a stirrer for 12 h at room temperature. Then, a solution containing the corresponding amounts of PG, ethanol, OA, and T80 was added to the aqueous dispersion of HPMC and, finally, NT was added to that mixture, which was continuously stirred for one additional hour.

### 2.5. In Vitro Skin Permeation

Rat full-thickness skin was obtained from previously sacrificed rats. Their hair was cut with an electric trimmer and pieces of skin (about 25 mm diameter and 1 mm thick) were mounted between the halves of vertical Franz-type diffusion cells, with the stratum corneum facing the donor chamber containing the gel with the NT. The receptor solution being in contact with the dermal side was composed of phosphate buffer saline pH 7.4 (9 mL). The receptor solution was continuously stirred with a small magnetic bar. The area available for diffusion was 1.45 cm^2^. NT gel (0.4 mL) was uniformly spread on the skin surface, the donor chamber was covered with parafilm^®^, and the cells were immersed in a water bath of 37 °C. To maintain infinite dose conditions, the gel was removed from the donor chamber and replaced with fresh gel of the same composition 9 h after starting the experiment. The receptor fluid was withdrawn (200 µL) and refilled with fresh medium at 3, 5, 7, 9, 24, 26, 28, and 30 h. The collected samples were analysed by HPLC [27] to determine the *Q_R_* values. Briefly, chromatographic separation was performed on a Waters Spherisorb ODS2 (4.6 × 250 mm) column at room temperature. The mobile phase consisted of a mixture of acetonitrile/water (85/15, *v*/*v*), delivered at a flow rate of 1 mL/min. Samples were derivatised with Fmoc-Cl and the fluorescence detector was set at excitation and emission wavelengths of 260 nm and 310 nm, respectively.

Equation (9) was fitted to the experimental data (*Q_R_*, *t*) using the NONMEM program (version 7.3; ICON Development Solutions, Ellicott City, MD, USA) to obtain the values of the parameters *R*_0_ and *k_out_*.

### 2.6. In Vivo Skin Permeation

To study the in vivo skin permeation of NT, five rats were cannulated in the jugular vein [28], the hair on their backs was cut with an electric trimmer, and a device consisting of a polypropylene hollow cylinder with a base area of 1.39 cm^2^ (exposure area) was fixed to their skin with cyanoacrylate adhesive [29]. NT gel (0.4 mL) was introduced in the cylinder and the open base was sealed with a small piece of rubber. Blood samples were collected through the jugular cannula at 8, 12, 24, 27, 30, 46, 48, 51, and 54 h using heparinized syringes as previously indicated, and the NT plasma concentrations were determined by HPLC [27]. At the end of the experiments, the rats were sacrificed and the amount of NT remaining in the gel, as well as the amount in the skin in contact with the gel, were quantified. The inside part of the device used for holding the gel was washed with saline to obtain a solution containing the unabsorbed NT. The skin was excised from the rats, cut in small fractions, and NT was extracted with a mixture of acetonitrile and water (50/50, *v*/*v*). The obtained NT plasma concentrations were plotted against the sampling times and the area under this curve extrapolated to time infinity (*AUC*) was determined using the trapezoidal rule [26]. The apparent clearance of the NT in rats administered with the formulated gel (*CL_app_*) was calculated as the quotient between the administered NT dose and the *AUC* value.

### 2.7. Simulation of NT Plasma Concentrations in Rats Dosed with the Formulated Gel

The plasma concentrations of NT were simulated in rats percutaneously dosed with the formulated NT gel using the pharmacokinetic parameters of NT in the rats administered intravenously, the value of the *k_out_* constant obtained in the in vitro skin permeation experiments, and a *k_in_* value estimated using Equation (12), the administered dose, and the amount of NT remaining in the gel determined in the in vivo skin permeation experiments. The pharmacokinetic parameters of NT in the rats administered intravenously were obtained by fitting, using the NONMEM, a two-compartment model to the plasma concentrations obtained in a previous study [27].

A graphic representation of the pharmacokinetic model used for the simulation of NT plasma concentrations is shown in Figure 3. This model contains three compartments, two of which (the central and peripheral compartments) have the same meaning as in the model fitted to the plasma concentrations obtained after the intravenous administration, while third represents the skin on which the formulation of the NT is applied in the in vivo skin permeation experiments. The differential equations defining the model and the parameter values obtained as described above were entered into NONMEM to predict the plasma concentrations, which were compared with the experimental results obtained in the in vivo skin permeation experiments.

## 3. Results

Figure 4 shows the mean cumulative amounts of NT in the receptor chamber of the diffusion cells obtained in the in vitro experiments, as well as the theoretical curve obtained after fitting Equation (9). The values obtained for *R*_0_ (or *J_ss_*) and *k_out_* were 229 ± 13 μg/cm^2^/h and 0.10 ± 0.01 h^−1^, respectively (value ± SE). Considering that the NT concentration in the donor chamber was 50,000 µg/mL, a value of *K_p_* = 0.0046 cm/h was obtained for the NT in rat skin when the formulated gel was applied.

The pharmacokinetic parameters of NT obtained by fitting a two-compartment model to the plasma concentrations obtained after the intravenous administration of 0.5 mg are shown in Table 1.

The average amount of NT remaining in the gel at the end of the in vivo skin permeation experiments (54 h) was 3% of the initial amount, which made it possible to estimate an in vivo *k_in_* value of 0.065 h^−1^ by means of Equation (14). Additionally, the *CL_app_* of the NT in these rats was 3126 mL/h; this value being higher than the *CL* value obtained in the rats administered intravenously (Table 1). To simulate the NT plasma concentrations in rats administered percutaneously with the formulated gel, the model in Figure 3 and the following values were used: the *k_out_* value obtained in vitro, the *k_in_* value obtained in vivo, the values of the distribution parameters obtained in rats administered intravenously (*V*_1_, *V_T_*, and *CL_ic_* in Table 1) and the *CL_app_* of the NT in rats administered percutaneously. Figure 5 shows the simulated NT concentrations, whose profiles were close to the experimental concentrations.

In order to evaluate the relevance of the parameter *k_out_*, an additional simulation was performed, which excluding this parameter in the model (i.e., using a two-compartment model with first-order absorption and the same parameters (*k_in_*, *V*_1_, *V_T_*, *CL_ic_* and *CL_app_*). This simulation provided an NT plasma profile far from the experimental concentrations (Figure 5), which indicates the usefulness of the *k_out_* parameter determined in vitro to predict plasma concentrations.

Equation (15) was used with the above values of *k_in_* and *k_out_* to predict the percentage of administered NT remaining in the skin of the rats at the end of the experiments (54 h), and a value of 4.7% was obtained. This predicted value was in good agreement with the value of 5.2% obtained experimentally.

## 4. Discussion

The transport of permeants through the skin has traditionally been described by equations derived from Fick’s laws of diffusion. Although these equations are based on very sound concepts, their complexity makes them difficult to apply to the analysis of data obtained from in vitro experiments using infinite or finite dose regimens. A simple equation describing the experimental data can only be obtained if a steady state in the diffusion process is reached during an infinite dose experiment (Equation (4)).

In this work, a simple one-compartment model has been used to describe the transport of a permeant across a membrane (skin), which provides relatively simple expressions for the amount of permeant in the membrane, the cumulative amount that crosses the membrane, and the flux of permeant, for both the infinite and the finite dose regimens. A limitation of the model and derived equations is that no delay in the amount of permeant crossing the membrane is considered, i.e., as the permeant reaches the membrane, a given fraction leaves it. However, an initial delay can easily be incorporated into the equations by replacing *t* with (*t* − *t*_0_), where *t*_0_ represents the time delay.

Some ideal physicochemical properties proposed for transdermal drug delivery are: a molecular weight of below 400 g/mol, a log *P*(o/w) in the −1 to 4 range, and a melting point of below 200 °C [30]. In the case of NT, the molecular weight is 263.4 g/mol, the log *P*(o/w) is 3.9–4.7, and the meting point is 213–215 °C (pubchem). These values are within or slightly outside the ideal ranges, which suggests a good percutaneous transport of this molecule. The in vitro evaluation of the formulated NT gel using rat skin showed a measurable transport of NT, with quantifiable concentrations in the receptor chamber 3 h after the start of the experiments. Fitting Equation (9) to the corresponding cumulative amounts of NT allows for estimating the value of *k_out_*, in addition to the parameters usually determined in infinite dose experiments (*J_ss_* and *K_p_*). In the proposed one-compartment model, the rate constant *k*_out_ has the same meaning for both the infinite and finite dose regimens (Figure 2), and, consequently, it can be used in combination with other parameters (*k_in_* and disposition parameters) to simulate the plasma concentrations of NT after the transdermal administration of a finite dose.

The transdermal administration of the formulated NT gel to rats resulted in plasma levels of approximately 150 ng/mL between 8 and 30 h post-administration. These levels are higher than the minimum concentration of 40 ng/mL recommended for smoking cessation therapy [21] and slightly higher than the upper limit of the therapeutic range for treatment of depression in humans (50–150 ng/mL) [19,20]. Although a direct extrapolation to humans is not feasible given the differences in skin permeability characteristics in both species, with rat skin being generally more permeable than human skin [31], the results obtained suggest that the NT formulation used in this study could provide therapeutic levels of NT in humans.

The apparent plasma clearance of NT in rats administered percutaneously (3126 mL/h) was higher than the plasma clearance determined in rats administered intravenously in a previous study (1580 mL/h). Although part of the discrepancy between both values could be attributed to metabolic degradation of NT in the skin, this difference is probably due to the fact that both studies were performed at different times, separated by several years, and reflects the fact that different populations of rats were used. In addition, the plasma levels of NT obtained with intravenous administration were higher than those obtained with transdermal administration, which could cause some degree of saturation in the elimination of NT and, consequently, a decrease in plasma clearance. The differences between rat populations are expected to have a greater impact on elimination (*CL*) than distribution (*V*_1_, *V_T_*, and *CL_ic_*) processes. Therefore, the values of the distribution parameters determined in rats from the previous study, in which intravenous administration was applied, were used in the simulations, but the *CL* value was replaced by the *Cl_app_* value obtained from the rats of this study. In order to use the model shown schematically in Figure 3 for the simulation of NT plasma concentrations, a value for *k_in_* is required, but unlike *k_out_*, its value cannot be estimated from the results obtained in vitro using an infinite dose regimen. However, this value can be easily estimated in vitro by means of Equation (14), if the amount of permeant remaining in the percutaneous formulation is determined at some time after the start of administration. Finally, a good agreement was obtained between the simulated and observed plasma concentrations of NT (Figure 5). On the other hand, the simulated concentrations did not reproduce the profile of the experimental ones when *k_out_* was not included in the model.

Compartmental models have previously been used to describe blood drug concentration–time profiles considering a constant flux of the drug through the skin or typical first-order absorption kinetics associated with a “flip-flop” phenomenon [13]. More complex compartmental models, with three or more rate constants, have also been proposed to describe the transport of drugs through the skin [12,13,14]. However, in this work, transport through the skin has been modelled using only two rate constants, which has simplified the pharmacokinetic model aimed at predicting plasma concentrations.

## 5. Conclusions

This work shows the potential usefulness of the *k_out_* parameter determined in vitro to predict plasma concentrations of drugs administered percutaneously. To do so, the differential Equations (12) and (13) or the integrated Equation (16), which describe drug entry into systemic circulation, can be combined with the differential equations of a typical two-compartment model that describes the drug disposition. Nevertheless, further studies with other molecules are needed to assess the general validity of the described approach.

## Figures and Tables

**Figure 1 pharmaceutics-14-01457-f001:**
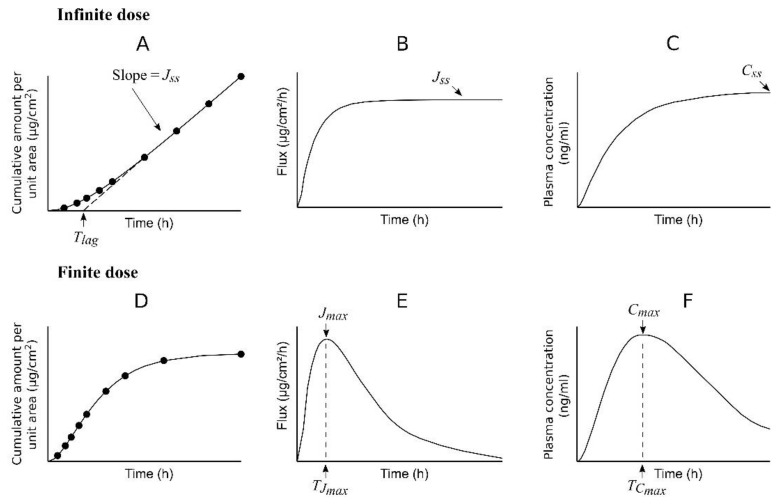
Graphs showing the different time profiles obtained with infinite (**A**–**C**) and finite (**D**–**F**) dose regimens. (**A**,**D**) cumulative amount of permeant in the receptor chamber, B and E: permeant flux, (**C**,**F**) plasma concentration of permeant.

**Figure 2 pharmaceutics-14-01457-f002:**
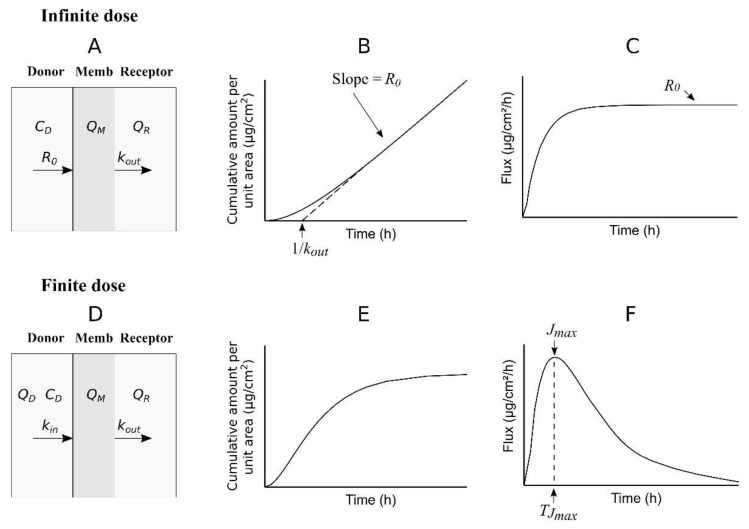
Schematic diagram of the one-compartment model used to describe the transport of a permeant across a membrane under infinite (**A**) and finite (**D**) dose conditions. The expected time profiles of the cumulative amounts of permeant in the receptor chamber and of the permeant flux are shown in graphs (**B**,**C**,**E**,**F**).

**Figure 3 pharmaceutics-14-01457-f003:**
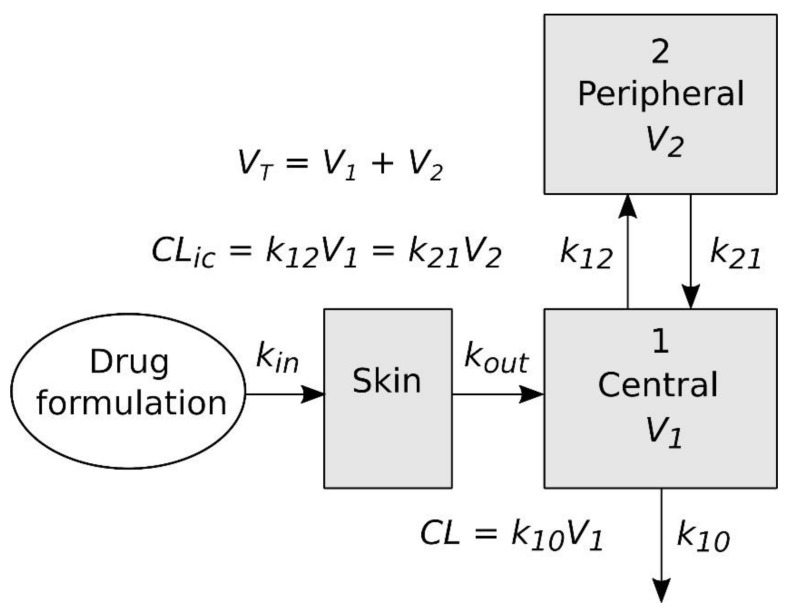
Schematic diagram of the three-compartment model used to simulate NT plasma concentrations in rats dosed percutaneously with the formulated NT gel. *CL*: total plasma clearance, *V*_1_: volume of the central compartment, *V*_2_: volume of the peripheral compartment, *V_T_*: total volume of distribution, *CL_ic_*: intercompartmental clearance, *k*_12_ and *k*_21_: first-order rate constants for mass transfer between the central and the peripheral compartments, *k*_10_: elimination first-order rate constant.

**Figure 4 pharmaceutics-14-01457-f004:**
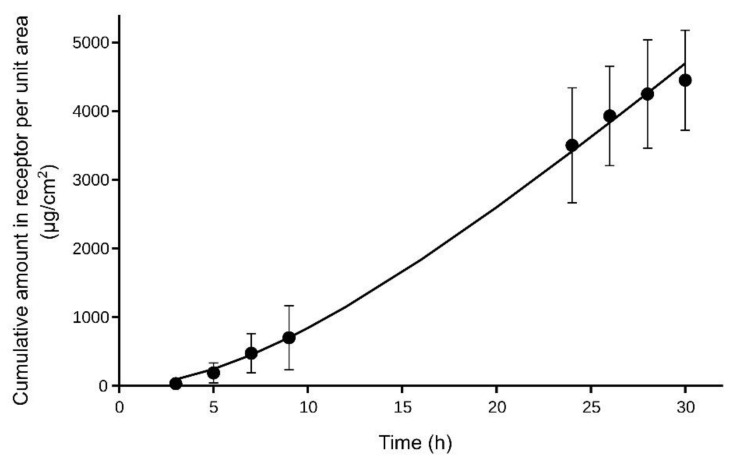
Cumulative amounts of NT (mean ± SD, *n* = 4) in the receptor chamber of the diffusion cells used in the in vitro skin permeation study.

**Figure 5 pharmaceutics-14-01457-f005:**
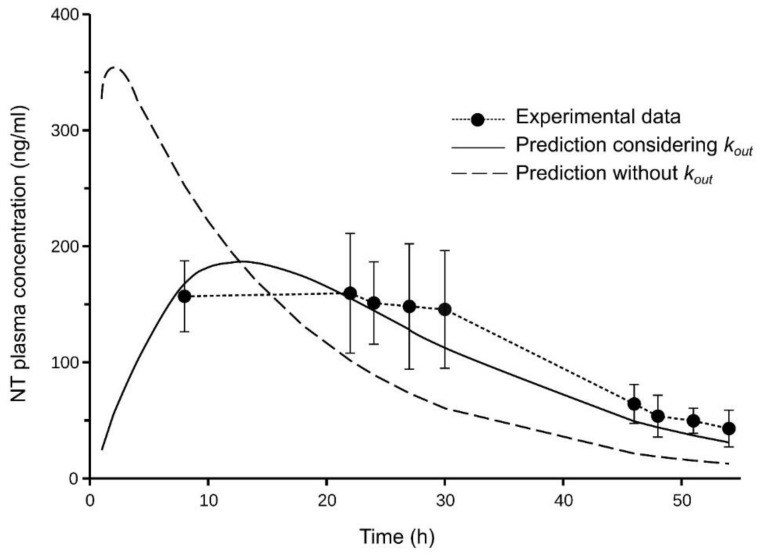
Plasma concentrations of NT (mean ± SD, *n* = 5) obtained from rats dosed with the NT gel. The continuous curve shows the predicted plasma concentrations using the three-compartment model depicted in Figure 3. The dashed curve shows the predicted concentrations when *k_out_* was not considered in the model.

**Table 1 pharmaceutics-14-01457-t001:** Pharmacokinetic parameters (two-compartment model) obtained in rats dosed intravenously with NT.

Parameter	Value ± SE
*CL* (mL/h)	1580 ± 282
*V*_1_ (mL)	488 ± 65
*V_T_* (mL)	1568 ± 160
*CL_ic_* (mL/h)	3340 ± 933

*CL*: total plasma clearance, *V*_1_: volume of the central compartment, *V_T_*: total volume of distribution, *CL_ic_*: intercompartmental clearance.

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
