# Peer review of "The Usefulness of In Vitro Percutaneous Absorption Experiments Applying the Infinite Dose Technique to Predict In Vivo Plasma Levels: Comparison of Model-Predicted and Observed Plasma Concentrations of Nortriptyline in Rats"

_pharmaceutics, 2022, doi:10.3390/pharmaceutics14071457_

Round 1

Reviewer 1 Report

Physicochemical properties of NT should be discussed and considered in this work, in the context of skin absorption and distribution given that NT is lipophilic with Ko/w higher than 4.

PBPK modeling and some physiological considerations would benefit this manuscript greatly. Did the authors consider blood flow on skin concentration and penetration rates in their model, or they considered it to be zero? This should be elaborated in the manuscript.

HPLC conditions need to be added in the methods section. 

The Animal Study Protocol number and approving institution should be added to the Material section, and in the IRB section of the manuscript, as per journal guidelines: https://www.mdpi.com/journal/pharmaceutics/instructions

Figure 5 needs a figure legend for 3 different lines.

Line 43, 47 in vitro and in vivo should be italic. 

What are some of the limitation of this study?

Author Response

Question 1: Physicochemical properties of NT should be discussed and considered in this work, in the context of skin absorption and distribution given that NT is lipophilic with Ko/w higher than 4.

Authors’ reply:

Several physicochemical properties of NT have been included in the revised manuscript and are discussed in the context of transdermal drug delivering.

Question 2: PBPK modeling and some physiological considerations would benefit this manuscript greatly. Did the authors consider blood flow on skin concentration and penetration rates in their model, or they considered it to be zero? This should be elaborated in the manuscript.

Authors’ reply:

The model presented in this paper is a compartmental kinetic model. It is not related to physiologically based pharmacokinetic (PBPK) modelling.  The kout constant determines the diffusion of drug molecules from the skin to the receptor chamber (in vitro) or to the blood (in vivo). The receptor solution was continuously stirred to avoid a relatively high drug concentration near the skin and the in vivo model assumes that blood flow is fast enough to rapidly distribute the absorbed molecules; that is, the blood flow does not limit the diffusion of the drug from the skin to the blood.

Question 3: HPLC conditions need to be added in the methods section.

Authors’ reply:

The HPLC conditions have been added in the methods section.

Question 4: The Animal Study Protocol number and approving institution should be added to the Material section, and in the IRB section of the manuscript, as per journal guidelines: https://www.mdpi.com/journal/pharmaceutics/instructions

Authors’ reply:

The animal study protocol number and approving institution have been added to section 2.3.

Question 5: Figure 5 needs a figure legend for 3 different lines.

Authors’ reply:

The required legend has been added to Figure 5.

Question 6: Line 43, 47 in vitro and in vivo should be italic.

Authors’ reply:

“In vitro” and “in vivo” on lines 43 and 47 are italicized in the revised manuscript.

Question 7: What are some of the limitations of this study?

Authors’ reply:

In the revised manuscript, a limitation of the study has been added.

Reviewer 2 Report

Dear authors,

In general, the work is nice and try to mechanistically show the in vivo pharmacokinetic behaviour after dermal administration. The topic is interesting for skin researchers.

I would appreciate to answer some points before publication.

The equations that predict the intra dermal and plasma drug levels after dermal application has been classically discussed. Please increase the state of the art and please include the following references and discuss accordingly.

Percutaneous Absorption. Drugs–Cosmetics–Mechanisms–Methodology, (ISBN: 1-57444-869-2)

Predictive Methods in Percutaneous Absorption (ISBN 978-3-662-47370-2)

Dermal Absorption and Toxicity Assessment (ISBN: 0-8493-7591-6)

Figure 2 is a little bit repetitive when looking at figure 1.

Also, section 2.1. is usually explained in several article and books. In my opinion should be referred in the introduction as the state of the art of the field.

Section 2.1.1. According to the figure 1C, at infinite dose a steady state plasma level is obtained. This model describes a constant entry from the skin to the systemic circulation (it should be described as zero order), but authors in equations of this section described an order one. Please clarify. Is the same case as transdermal patches that the entry drug level is a constant rate.

Section 2.5. The most common way to maintain infinite dose is to dose a higher dose in the donor compartment instead of substitution with fresh formulation. Why the authors chose this technique that involve more manipulation and more possibilities to damage the thin rat skin. In addition, by authors chose the point of 9 h to replace the dose on the donor compartment?

Section 2.6. Please specify the sampling times of in vivo study. How many animals were used? Why did authors not use compartmental analysis with in vivo samples using NONMEM? It seems that only simulations were carried out (section 2.7).

A clear conclusion about the equations that should be used to predict in vivo levels based on in vitro permeation test should be stated, according to the aims described by the authors.

Author Response

Question 1: The equations that predict the intra dermal and plasma drug levels after dermal application has been classically discussed. Please increase the state of the art and please include the following references and discuss accordingly.

Percutaneous Absorption. Drugs–Cosmetics–Mechanisms–Methodology, (ISBN: 1-57444-869-2)

Predictive Methods in Percutaneous Absorption (ISBN 978-3-662-47370-2)

Dermal Absorption and Toxicity Assessment (ISBN: 0-8493-7591-6)

Authors’ reply:

Paragraphs including these references have been added to the introduction and discussion sections of the revised manuscript.

Question 2: Figure 2 is a little bit repetitive when looking at figure 1.

Authors’ reply:

Figures 2A and 2D represent the one-compartment model used to describe the transport of a permeant across a membrane. Figures 2B, 2C, 2E and 2F show the graphs that will theoretically be obtained with the proposed model. These four plots are similar to figures 1A, 1B, 1D and 1E, indicating that the proposed model will provide plots similar to those obtained with the diffusion equations. Additionally, figures 2B and 2C show the new parameters Ro and kout. Therefore, the authors consider that these figures will be of help for the readers of the article.

Question 3: Also, section 2.1. is usually explained in several article and books. In my opinion should be referred in the introduction as the state of the art of the field.

Authors’ reply:

The authors agree that some equations have been described previously, but there are others, such as equations 9 to 11 and equation 17, which are used for the first time to describe the cumulative amount in the receptor (QR) and the flux (J), to the best of the authors' knowledge.

Question 4: Section 2.1.1. According to the figure 1C, at infinite dose a steady state plasma level is obtained. This model describes a constant entry from the skin to the systemic circulation (it should be described as zero order), but authors in equations of this section described an order one. Please clarify. Is the same case as transdermal patches that the entry drug level is a constant rate.

Authors’ reply:

Figure 2A shows that the proposed model contains a zero-order constant (R0) and a first-order constant (kout). This means that the permeant enters the membrane at a constant rate (R0) and leaves it according to first-order kinetics. Figures 1B, 1C and 2B show a pseudo-steady state at some time after the start of the experiment when an infinite dose regimen is used. The “constant entry from the skin to the systemic circulation” indicated by the reviewer is obtained at that pseudo-steady state. As Figures 1A and 2B show, a “constant entry” or a constant flux (Jss) is reached at some time after the start of the experiment. The entrance of the drug from the skin to the systemic circulation is kinetically represented by the product QM.kout (first-order kinetics). After the start of the experiment, QM increases and the entrance rate (QM.kout) also increases. However, after some time, a maximum and constant QM value will be reached (considering an infinite dose regimen) and, consequently, the entrance rate will also be constant. When this occurs, the flux is Jss and the entrance rate is equal to R0 (QM.kout = R0). This is what happens when a patch that theoretically provides a constant-rate release is used; the entrance of drug into the skin follows zero-orden kinetics (constant rate) but the entrance of drug into the systemic circulation is not constant at the beginning. A constant entry into the systemic circulation (constant flux, Jss) is only reached after some time.

Question 5: Section 2.5. The most common way to maintain infinite dose is to dose a higher dose in the donor compartment instead of substitution with fresh formulation. Why the authors chose this technique that involve more manipulation and more possibilities to damage the thin rat skin. In addition, by authors chose the point of 9 h to replace the dose on the donor compartment?

Authors’ reply:

As the reviewer indicates, an infinite dose regimen can be obtained using a high dose in the donor chamber. However, we prepared a 5 % drug formulation for the in vivo studies and we used the same formulation for the in vitro studies. After 9 h, the amount of drug in the receptor chamber was approximately 7 % of the initial amount, which indicates a decrease in the initial concentration in the donor chamber larger than 7 % (considering that some amount was also in the skin in addition to the amount in the receptor). Furthermore, this sampling time (9 h) was immediately before a relatively long time without sampling (overnight period).

Question 6: Section 2.6. Please specify the sampling times of in vivo study. How many animals were used? Why did authors not use compartmental analysis with in vivo samples using NONMEM? It seems that only simulations were carried out (section 2.7).

Authors’ reply:

The sampling times have been added in the revised manuscript.

The number of animals was indicated in Figure 5 (n = 5). This number has also been added in section 2.6 of the revised manuscript.

Fitting a compartmental model to the experimental concentrations allows to estimate the values of pharmacokinetic parameters. However, the aim of the work was to evaluate the usefulness of the kout value obtained in vitro to predict in vivo concentrations. For this reason, NONMEM was used to simulate NT plasma concentrations using the kout value obtained in vitro, which were compared to the observed plasma concentrations.

Question 7: A clear conclusion about the equations that should be used to predict in vivo levels based on in vitro permeation test should be stated, according to the aims described by the authors.

Authors’ reply:

The equations to be used to predict in vivo levels are the differential equations 12 and 13 or the integrated equation 16, which describe drug entry into the systemic circulation, combined with the differential equations of a typical two-compartment model that describes the drug disposition (Figure 3).

Reviewer 3 Report

In this study, the authors investigated the usefulness of in vitro percutaneous absorption experiments applying the infinite dose technique to predict in vivo plasma levels. Although the concept of this study is interesting, however, the provided data seems to be insufficient to confirm the authors’ concept.  The authors are recommended to respond to the following concerns

1- Many kinetic equations were introduced in the Introduction and Methodology sections, however, their connection in the results section was lacking.

2- Materials and methods section

- Section 2.5.: More details about skin permeation study should be provided such as thickness of skin, volume of receptor medium and how many mg of NT was in the gel applied to the donner compartment

- Section 2.6: It is preferable to provide a cartoon depicting the in vivo application of NT to rats

3- Results section needs more clarification about the data obtained from in vitro percutaneous absorption experiments and its correlation with the in vivo results. In addition, ambiguous descriptions were observed such as

- In line 316, the authors calculated Kp on the basis that NT concentration in the donor chamber was 50 mg, while in line 324, the authors tracked PK parameters following IV administration of only 0.5 mg. Why the drug concentrations applied topically and intravenously were different

- Line 336, the authors have to revise their description “Additionally, the CLapp of NT in these rats was 3,126 ml/h; this value being higher than the CL value obtained in rats ad- ministered intravenously (Table 1)”, where CLapp seems to be lower not higher.

4. Finally, the discussion section sounds to be a repetition of results description. More in depth discussion should be included to emphasize the usefulness of in vitro percutaneous absorption experiments applying the infinite dose technique to predict in vivo plasma levels.

Author Response

Question 1: Many kinetic equations were introduced in the Introduction and Methodology sections, however, their connection in the results section was lacking.

Authors’ reply:

Equations in the Introduction section show the complexity of the equations that describe diffusion of molecules across membranes. In the Materials and Methods section, the equations derived from a one-compartment model to describe the time course of a permeant in the membrane and in the receptor chamber are given. The authors consider that all of these equations are important to understand the background of the present study. Equation 9 is used to obtain the value of kout, equation 10 (compartmental approach) is equivalent to equation 4 (diffusional analysis). Equations 12 and 13 are used for the in vivo prediction. Equation 14 is used to estimate the value of kin. Equation 15 is used to obtain the amount of NT in the skin at the end of the in vivo experiments.

Question 2: Section 2.5.: More details about skin permeation study should be provided such as thickness of skin, volume of receptor medium and how many mg of NT was in the gel applied to the donner compartment

Authors’ reply:

The thickness of the skin and the volume of the receptor are given in the revised manuscript. The milligrams of NT in the gel applied to the donor chamber were indirectly given in the original manuscript: 0.4 ml of 5 % gel = 0.02 g = 20 mg.

Question 3: Section 2.6: It is preferable to provide a cartoon depicting the in vivo application of NT to rats

Authors’ reply:

A reference has been added showing a device similar to the one used in this study.

Question 4: In line 316, the authors calculated Kp on the basis that NT concentration in the donor chamber was 50 mg, while in line 324, the authors tracked PK parameters following IV administration of only 0.5 mg. Why the drug concentrations applied topically and intravenously were different

Authors’ reply:

Intravenous administration of any drug gives plasma concentrations much higher than extravascular (percutaneous) administration of the same dose. The intravenous lethal dose 50% (LD50) of NT in rats is 22.3 mg/kg (6.7 mg/rat). That is, the intravenous administration of the same dose used for transdermal absorption (20 mg/rat) would have caused the death of the animals. For this reason, the intravenous dose was lower than the dose applied on the skin.

Question 5: Line 336, the authors have to revise their description “Additionally, the CLapp of NT in these rats was 3,126 ml/h; this value being higher than the CL value obtained in rats administered intravenously (Table 1)”, where CLapp seems to be lower not higher.

Authors’ reply:

Clapp = 3,126 ml/h

CL = 1,580 ml/h

CLic = 3,340 ml/h

The sentence “Additionally, the CLapp of NT in these rats was 3,126 ml/h; this value being higher than the CL value obtained in rats administered intravenously (Table 1)” is correct.

Question 6: Finally, the discussion section sounds to be a repetition of results description. More in depth discussion should be included to emphasize the usefulness of in vitro percutaneous absorption experiments applying the infinite dose technique to predict in vivo plasma levels.

Authors’ reply:

New paragraphs have been added to the discussion section to emphasize the interest of the study.

Reviewer 4 Report

The authors have well-written manuscript regarding the evaluation of feasibility of Nortriptyline formulation for transdermal administration and to assess the usefulness of a kinetic parameter  estimated using the in vitro infinite dose technique to predict in vivo plasma levels when used in combination with pharmacokinetic parameters. The article is well designed and the claims are supported by the results. I hope the article will be interesting to scientific community and adds sufficient value to the literature. I may pasting here the minor comments that the authors can follow to make it easier for the readers to follow it.

  1. English language should be improved.
  2. The authors should mention the BCS class of Nortriptyline. There should be some more details about the model drug in the introduction section.
  3. Authors give us idea about the use of Nortriptyline as smoking cessation drug if it is administered transdermally. What about the indicated/labelled use of the drug (antidepressant)??? Whether these effects are affected by this infinite dose????
  4. Authors should elaborate the procedure for formulation development i.e. at what temperature it was made and for how much time the mixture was stirred to make the gel?
  5. The formulation was meant to be administered onto skin but in vitro permeation study was done on 37ºC. What is the scientific justification for conducting the study at 37ºC (“the cells were immersed in a water bath at 37 °C”). As the simulation conditions for the skin should have been maintained at 32ºC. Please explain it??

Author Response

Question 1: The authors should mention the BCS class of Nortriptyline. There should be some more details about the model drug in the introduction section.

Authors’ reply:

The BCS class of Nortriptyline has been included in the revised manuscript.

Question 2: Authors give us idea about the use of Nortriptyline as smoking cessation drug if it is administered transdermally. What about the indicated/labelled use of the drug (antidepressant)??? Whether these effects are affected by this infinite dose????

Authors’ reply:

In the manuscript, it is indicated that “These levels are higher than the minimum concentration of 40 ng/ml recommended for smoking cessation therapy and slightly higher than the upper limit of the therapeutic range for treatment of depression in humans (50 – 150 ng/ml).”

Question 3: Authors should elaborate the procedure for formulation development i.e. at what temperature it was made and for how much time the mixture was stirred to make the gel?

Authors’ reply:

The procedure used to obtain the NT gel, including temperature and stirring time, is described in the revised manuscript.

Question 4: The formulation was meant to be administered onto skin but in vitro permeation study was done on 37ºC. What is the scientific justification for conducting the study at 37ºC (“the cells were immersed in a water bath at 37 °C”). As the simulation conditions for the skin should have been maintained at 32ºC. Please explain it??

Authors’ reply:

The temperature at which an in vitro percutaneous penetration experiment is conducted is normally 37 ºC, maintained by the use of circulating water. This results in a temperature of about 32 ºC at the skin surface, which is in contact with the environmental air (See reference 12, page 36).

Round 2

Reviewer 1 Report

Thank you for addressing all of my questions and comments. In my opinion the manuscript can be accepted. 

Author Response

Thank you very much.

Author Response

Question 3: Also, section 2.1. is usually explained in several article and books. In my opinion should be referred in the introduction as the state of the art of the field.

Authors’ reply:

The authors agree that some equations have been described previously, but there are others, such as equations 9 to 11 and equation 17, which are used for the first time to describe the cumulative amount in the receptor (QR) and the flux (J), to the best of the authors' knowledge.

If the equations were previously reported by other authors, they should be referred. Please include references in the equations from other authors.

With regards to equations in section 2.1., equations 7 and 8 were obtained from reference 26, as indicated in the manuscript. Equations 9 to 11 are used for the first time to describe the cumulative amount in the receptor (QR) and the flux (J), in an infinite dose regimen, to the best of the authors' knowledge. Equations 12-15 were obtained from reference 26, as indicated in the manuscript. Equations 16-18 are used for the first time to describe the cumulative amount in the receptor (QR), the flux (J) and the time to achieve Jmax (TJmax) in a finite dose regimen, to the best of the authors' knowledge.

Question 7: A clear conclusion about the equations that should be used to predict in vivo levels based on in vitro permeation test should be stated, according to the aims described by the authors.

Authors’ reply:

The equations to be used to predict in vivo levels are the differential equations 12 and 13 or the integrated equation 16, which describe drug entry into the systemic circulation, combined with the differential equations of a typical two-compartment model that describes the drug disposition (Figure 3).

That’s fine, but I recommend to include this conclusion that authors reported as reviewer’s answer in a specific conclusion section at the end of the article.

A conclusion section has been included in the revised manuscript including the information reported as reviewer’s answer.

Reviewer 3 Report

The authors have replied all the concerns raised by the reviewer.

Author Response

Thank you very much.